# Postural Stability in Athletes: The Role of Age, Sex, Performance Level, and Athlete Shoe Features

**DOI:** 10.3390/sports8060089

**Published:** 2020-06-17

**Authors:** Albina Andreeva, Andrey Melnikov, Dmitry Skvortsov, Kadriya Akhmerova, Alexander Vavaev, Andrey Golov, Viktorya Draugelite, Roman Nikolaev, Serafima Chechelnickaia, Daria Zhuk, Alexandra Bayerbakh, Vladislav Nikulin, Erika Zemková

**Affiliations:** 1Department of Sports Biomechanics, Moscow Center of Advanced Sport Technologies, 129272 Moscow, Russia; skvortsov.biom@gmail.com (D.S.); cstsk@mos.ru (K.A.); vavaev@gmail.com (A.V.); golov.andrey@hotmail.com (A.G.); draugelite@ya.ru (V.D.); 2Department of Physiology, Institute of Tourism, Recreation, Rehabilitation and Fitness, Russian State University of Physical Education, Sport, Youth and Tourism, 105122 Moscow, Russia; meln1974@yandex.ru; 3Clinical Rehabilitation Research Center for Patients in Remission “Russkoye Pole” under Dmitry Rogachev National Research Center of Pediatric Hematology, Oncology and Immunology, 117198 Moscow, Russia; dar-2006@bk.ru (S.C.); dashkazhuk96@mail.ru (D.Z.); baer83@mail.ru (A.B.); vladavlad94@mail.ru (V.N.); 4Physical Culture Department, P.A. Solovyov Rybinsk State Aviation Technical University, 152934 Rybinsk, Russia; nikolaev.r.u@yandex.ru; 5Department of Biological and Medical Sciences, Faculty of Physical Education and Sports, Comenius University in Bratislava, 814 69 Bratislava, Slovakia; erika.zemkova@uniba.sk; 6Sports Technology Institute, Faculty of Electrical Engineering and Information Technology, Slovak University of Technology, 811 07 Bratislava, Slovakia

**Keywords:** postural stability, youth athletes, sex, stabilometry, performance level

## Abstract

The effects of different factors—such as age, sex, performance level, and athletic shoe features—on postural balance in athletes remain unclear. The main objective of our study is to identify the features of postural stability in athletes of different age, sex, performance level, and using different types of athletic shoes. This study assessed postural stability in athletes (*n* = 936, 6–47 years) in a normal bipedal stance with eyes open (EO) and eyes closed (EC). Postural stability was evaluated based on the center of pressure (COP), sway area (AS), and velocity (VCP) while standing on a stabiloplatform. Children (6–12 years) and teen athletes (13–17 years) showed reduced AS-EO (*p* < 0.01) and VCP-EO (*p* < 0.01) compared to control (*n* = 225, 7–30 years). In male and female athletes aged 18+, only VCP-EC was lower versus control. In females (13–17 and 18+), VCP-EO and EC were lower than in males (*p* < 0.05). Only in the Shooting group, the athletes’ performance levels had an effect on VCP-EO (*p* = 0.020). Long use of rigid athletic shoes with stiff ankle support was associated with reduced posture stability. Postural stability in athletes was mostly influenced by the athlete‘s age, and, to a lesser extent, by their sex, performance level, and athlete shoe features.

## 1. Introduction

Good postural balance reduces the risk of sports injuries [1,2,3] and their negative consequences for the athlete’s physical condition and career. Good postural balance is a prerequisite for improving the control of voluntary movements in sports and, consequently, for enhancing athletic performance [4]. Postural balance has not been sufficiently studied on large samples of athletes [5] of different ages, sex, performance levels, and using different types of athlete shoes.

The bulk of the studies have evaluated postural balance in subjects under 12–13 years, and very few studies have compared it between adolescents and young adults [6,7]. It was shown that the center of pressure (COP) and head sway parameters in bipedal stance reach adult levels by the age of 15 years, after which they no longer differ from those in adults (20–28 years old). In males, however, the ability to maintain unipedal balance increases throughout the period of adolescence and improves at the late stage of maturation [8]. In boys, postural stability continued to improve at the ages of 9 to 16, whereas in girls, it approached the adult levels by the age of 10 [9].

Different authors link the higher postural stability of females to their earlier physical maturation [10], greater diligence and attention when doing postural tasks [11], lesser body weight [12], anatomical features (a lower center of gravity in teenage girls due to a relatively wider pelvis and narrower shoulders) [10], and a better trainability of postural regulation system [13].

Given the interdependence of motor and postural abilities [14], one can expect that a more skilled athlete would have higher postural stability. Indeed, some authors have shown that the COP fluctuations in shooters [15], football players [16], and rhythmic gymnasts [17] of expert or international performance level are higher than in their less-skilled counterparts. A recent systematic review of this issue showed a positive relationship between postural stability and the level of athletic performance [18]. On the other hand, a number of studies have failed to find a close linkage between the level of athletic performance and body vibrations in a normal vertical stance in gymnasts [19], acrobats [20], wrestlers [21], surfers [22], and other athletes. We can surmise and test a hypothesis that such a linkage would show up in sports associated with a high level of postural stability, and would be absent in athletes with a low ability to maintain a static posture.

Finally, different authors point to a surprisingly reduced postural stability in higher-level athletes in certain sports compared with the lower-level ones, e.g., in alpine skiers [23], acrobats [20], and surfers [22]. The studies indicate that there are some sports-specific conditions that can modify normal postural strategies for maintaining a vertical posture, resulting in lower postural stability in the standard test. Such sports-specific conditions may include the long regular use of rigid and high athletic shoes to help stabilize the ankles, maintain body balance on a sliding surface, and increase the speed of movement, for example, in skiing or speed skating. One can expect that chronic use (several hours daily) of such footwear will reduce postural stability under normal test conditions without shoes [23].

Thus, it remains unclear how postural balance in athletes is related to such factors as their age, sex, performance level, and the features of their athletic shoes.

In this study, we assessed postural stability during a bipedal stance under eyes-open (EO) and eyes-closed (EC) conditions on a large sample of athletes engaged in 41 sports and in non-athletes used as control. The main objective of our study is to identify the features of postural stability in a normal vertical stance in athletes of different ages, sex, performance levels, and with different types of athletic shoes. Our objectives are (1) to identify sex- and age-related differences in static balance between athletes and control, (2) to determine the relationship between postural stability and performance level in athletes with high and low postural stability, (3) to determine the relationship between postural stability and shoe features as a chronic factor that can limit ankle mobility and affect posture control.

## 2. Materials and Methods

Study management. All studies were performed at the Center of Sports Innovative Technologies and Training of National Teams under the Department of Physical Culture and Sports of the City of Moscow (MCAST) in 2014–2018. Tests were performed during preseason training sessions by three specialists from the MCAST. Eligible subjects were to have no musculoskeletal injuries or any other contraindications to the tests. Prior to participation, they signed an appropriate informed consent form.

The study was conducted in compliance with the principles of the Declaration of Helsinki (2013) and was approved by the Local Ethics Committee of the MCAST (Protocol 11 dated 9 December 2019).

Study subjects. The study enrolled 936 athletes from various sports and 225 non-athlete controls not systematically engaged in any sport. All athletes were divided into 13 groups (Table 1, Figure 1) based on similarities in sports techniques.

The inclusion criteria were as follows: at least 8 h of sports practice per week; more than 2 years of sports practice for children, and more than 3 years for teenagers and adults. The exclusion criterion was the presence of any musculoskeletal or nervous system disorder that could affect postural stability. The non-athlete controls were students from secondary and higher educational establishments, who were not systematically engaged in sports activities (<3 times a week).

Tests and assessments. The body weight and height of the subjects were measured using conventional methods before assessing their postural balance. Postural stability was assessed using a force platform Stabilan-01-2^®^ (ZAO OKB “Ritm”, Taganrog, Russia). Data from the force platform were sampled at 50 Hz and filtered by two analog low-pass filters with bandwidths of 7 and 15 Hz [24,25,26,27]. Then, the signal was filtered by the analog-to-digital converter using the third-order Sinc filter. We assessed the postural stability of the subjects during a bipedal stance under the following conditions: (1) with EO, gaze directed to a black circle at a 2-m distance for 60 s; (2) with EC, for 60 s. The subjects were instructed to stand as still as possible on the force platform, with their heels 2 cm apart, the stance angle of 30 degrees, and their arms hanging loosely at the sides of the body. Under the eyes-open condition, the participants had no visual feedback about their CoP location. The subjects were asked to do some simple cognitive tasks during the test. Under the eyes-open condition, subjects counted the number of circles displayed. Under the eyes-closed condition, they counted the number of sound signals. The test of athletes was carried out in the morning before functional load or speed–strength testing. The following stabilometric parameters were recorded using the Stabilan 01-2 software program (ZAO OKB “Ritm”): mean velocity of the center of pressure displacement (VCP), mm/s, and area of statokinesiogram (AS), mm^2^ [27].

VCP and AS are the most commonly used direct stabilometric parameters [19,28]. Lower VCP and AS values reflect weaker muscular postural regulation and hence a higher efficiency of the postural balance [29].

Statistics. The analyses were performed using Statistica v.12 software (StatSoft, Inc., Tulsa, OK, USA). The normality of variable distribution was verified based on the Shapiro–Wilk test. Since the test variables had non-normal distribution, the correlation (Pearson correlation), regression, single-factor (ANOVA), and covariance (ANCOVA) analyses were performed on Box–Cox transformed variables. Athletes-versus-control comparisons, by age subgroups, were done using the unpaired Student’s *t*-test. ANCOVA evaluated the size of the effect of age on differences in the stabilographic parameters between athletes and controls (with age as the covariate).

Sex-related differences in the stabilometric parameters were determined: (1) within the overall group, regardless of age and sports engagement, (2) within age subgroups (children, teens, and adults), regardless of sports engagement and (3) within the group of athletes and the group of controls, taking account of age using a three-way analysis of variance. The independent factors were (1) sex (male, female), (2) age (children, adolescents, and adults), (3) sports engagement (athlete or non-athlete control). The dependent variables were stabilometric parameters: AS-EO, AS-EC, VCP-EO, VCP-EC.

A posthoc Tukey’s HSD test was applied for significant ANOVAs. The significance level α was set at 0.05. The effect size was calculated for each significant pairwise comparison. This effect size differs from that used to determine the sample size for the study, and it characterizes the magnitude of the calculated mean difference. According to Cohen’s (1988) criteria, *d* = 0.2 should be considered a “small” effect size, 0.5 represents a “medium” effect size, and 0.8 a “large” effect size.

## 3. Results

### 3.1. COP Sway Parameters in Athletes, by Age Group

Negative correlations were noted between age and the stabilographic variables under study: *r* = −0.40 for VCP-EO (*p* < 0.0001), *r* = −0.29 for VCP-EC (*p* < 0.0001), *r* = −0.28 for AS-EO (*p* < 0.0001), and *r* = −0.18 for AS-EC (*p* < 0.00001). Therefore, all athletes were divided into three age groups: children (under 12 years of age, inclusive), teens (13–17 years old), and adults (18 years and older). The stabilographic parameters were compared between athletes and non-athletes within each age group. In all age groups, a simple pairwise comparison showed (Table 2) that VCP-EO (*p* < 0.01 for all age groups; *d* = 0.90, *d* = 0.94, and *d* = 0.32 for children, teenagers, and adults, respectively) and VCP-EC (*p* = 0.022, *d* = 0.31 for teenage boys; *p* = 0.002, *d* = 0.43 for teenagers, *p* = 0.002, *d* = 0.44 for adults) in athletes were lower than in controls. AS-EO was lower in teenage male athletes (*p* = 0.0001, *d* = 0.63) and teenage athletes (*p* = 0.001, *d* = 0.46), whereas AS-EC was only lower in teenage male athletes (*p* = 0.019, *d* = 0.31). In adult athletes, AS-EO and AS-EC did not differ from control. However, because the mean age of athletes in all age subgroups was slightly, though significantly, higher than of respective non-athlete controls, we performed an ANCOVA, with age as the covariate. According to the ANCOVA for the eye-open tests, VCP-EO and AS-EO remained reduced in the subgroups of teenage male athletes and teenage athletes (*p* < 0.01) but did not differ from control in the adult subgroup of athletes (*p* > 0.07). In the eyes-closed test, VCP-EC in teenage males did not differ from control (*p* = 0.242), though it did differ in teenage (*p* = 0.012) and adult athletes (*p* = 0.003).

### 3.2. Sex-Related Differences in COP Sway Parameters in Athletes

Three-factor ANOVA (sex (2: male, female), engagement in sports (2: athlete, control), and age (3: children under 12 inclusive, teenagers of 13–17, adults of 18 and older) was used to evaluate sex-related differences in dependent COP sway parameters in the total group of all study subjects, in age subgroups, and in subgroups based on engagement or non-engagement in sports (athletes and non-athlete controls).

#### 3.2.1. AS-EO

Neither sex (F(1, 1149) = 1.10, *p* = 0.30), nor the interaction between sex and age (F(2, 1149) = 0.34, *p* = 0.712) or between sex, age, and sports engagement (F(2, 1149) = 1.06, *p* = 0.35) had any effect on AS-EO.

#### 3.2.2. AS-EC

In the total group, AS-EC in females was lower than in males (F(1, 1149) = 5.7278, *p* = 0.017). Neither the interaction between sex and age (F(2, 1149) = 0.38, *p* = 0.683) nor the interaction between sex, age and sports engagement (F(2, 1149) = 2.44, *p* = 0.088) had any effect on AS-EC.

#### 3.2.3. VCP-EO

Neither sex (F (1, 1149) = 0.39, *p* = 0.532) nor the interaction between sex and age (F (2, 1149) = 1.0, *p* = 0.365) had any effect on VCP-EO, yet the interaction between sex, age, and sports engagement (F(2, 1149) = 4.73, *p* = 0.009) influenced VCP-EO (Figure 2). While teenage female (Tukey’s HSD test *p* < 0.01) and adult female athletes had lower VCP-EOs (Tukey HSD test *p* < 0.01) than the respective male athletes, adult female non-athletes had a higher VCP-EO than adult male non-athletes (Tukey’s HSD test *p* < 0.01).

#### 3.2.4. VCP-EC

In the total group, VCP-EC in females was lower than in males (F(1, 1149) = 4.52, *p* = 0.034). The interaction between sex and age had no effect on VCP-EC (F(2, 1149) = 0.51324, *p* = 0.59869). The interaction of sex, age, and sports engagement was shown to have a significant effect on VCP-EC (F (2, 1149) = 4.04, *p* = 0.018; see Figure 3). In the athlete group, teenage females (Tukey’s HSD test *p* = 0.013) and adult females (Tukey’s HSD test *p* < 0.001) had lower VCP-EC’s compared with males. VCP-EO did not differ between males and females in the control group.

### 3.3. COP Sway Specifics Associated with the Type of Sports Footwear Used by Athletes

All athletes were divided into 3 groups based on the features of their sports shoes. The first group included athletes who trained barefoot or in leather-sole shoes, such as wrestling shoes (Without group, *n* = 266): Boxing (*n* = 38), Karate (*n* = 9), Kickboxing (*n* = 9), Thai Boxing (*n* = 10), Taekwondo (*n* = 40), Free-Style Wrestling (*n* = 8), Greco Roman (*n* = 29), Judo (*n* = 16), Sumo (*n* = 18), Gymnastics (*n* = 25), Rhythmic Gymnastics (*n* = 34), and Trampoline Jumping (*n* = 30). The second group included athletes who trained in soft athletic shoes, with elastic soles and without ankle support, such as running shoes, boots (Shoes group, *n* = 376): Basketball (*n* = 73), Handball (*n* = 17), Practical Shooting (*n* = 10), Archery (*n* = 7), Trap Shooting (*n* = 9), Tennis (*n* = 43), Table Tennis (*n* = 14), Football (*n* = 86), Curling (*n* = 17), Sprinting (*n* = 4), Styer Running (*n* = 7), Orienteering (*n* = 34), Cheerleading (*n* = 27), Climbing (*n* = 20), High Jumping (*n* = 10), and Skeleton (*n* = 14). The third group included athletes training in rigid shoes with ankle support (Hard Shoes group, *n* = 262): Biathlon (*n* = 46), Alpine Skiing (*n* = 32), Snowboarding (*n* = 17), Figure Skating (*n* = 80), Freestyle (*n* = 7), Speed Skating (*n* = 6), Hockey (*n* = 18), Short Track (*n* = 20), and Cross-Country Skiing (*n* = 37). The following sports were excluded from analysis: Rowing (*n* = 6), Canoeing (*n* = 4), Canoe Slalom (*n* = 4), and Sailing (*n* = 8). ANOVA did not reveal any differences in AS-EO (F(2, 878) = 0.60, *p* = 0.55) and AS-EC (F(2, 878) = 0.31, *p* = 0.73) among the shoe feature-based groups. However, VCP-EO and VCP-EC were significantly higher in the Hard Shoes group than in the Shoes group (ANOVA: F(2, 878) = 8.22, *p* = 0.0003; posthoc Tukey’s HSD test *p* = 0.0002; Figure 4A). There was also a trend towards a VCP-EC increase in the Hard Shoes group compared with the Without group (Tukey’s HSD test *p* = 0.066; Figure 4B).

### 3.4. Relationship between COP Sway Parameters and Sports Performance Level

Sports qualification ranks (according to the Russian Unified Sports Classification System) were available for 118 child athletes (6–12 years old). Regardless of the practiced sport, they were divided into 3 groups based on qualification rank (i.e., sports performance level): (1) First- to Third-Class Junior Sportsmen (*n* = 25); (2) First- to Third-Class Adult Sportsmen (*n* = 77), and (3) Candidates Master of Sports (*n* = 16). EO and EC postural stability tests found no correlation for any of the COP variables: VCP-EO (ANOVA *p* = 0.98), VCP-EC (ANOVA *p* = 0.44), AS-EO (ANOVA *p* = 0.89), and AS-EC (ANOVA *p* = 0.34). Thus, there was no relationship between postural stability and sports performance level in this age group.

In the teenage athlete group (13–17 years old), sports qualification ranks were available for 498 subjects. Regardless of the practiced sport, the subjects were divided into 4 groups based on their qualification rank: (1) First- to Third-Class Junior Sportsmen (*n* = 20); (2) First- to Third-Class Adult Sportsmen (*n* = 227); (3) Candidates Master of Sports (*n* = 221); (4) Masters of Sports (*n* = 30). ANOVA failed to reveal any significant differences in the stabilographic variables among the performance level-based groups: VCP-EO (ANOVA *p* = 0.147), VCP-EC (ANOVA *p* = 0.146), AS-EO (ANOVA *p* = 0.062), and AS-EC (ANOVA *p* = 0.075).

In the adult athlete group (aged 18 and older), sports qualification ranks were available for 228 subjects. Based on qualification rank, the subjects were allocated to 4 groups, regardless of sports discipline: (1) First- and Second-Class Adult Sportsmen (*n* = 22); (2) Candidates Master of Sports (*n* = 93); (3) Masters of Sports (*n* = 81); (4) International Class Masters of Sports (*n* = 32). ANOVA found no significant differences among the performance level-based groups of athletes in terms of VCP-EO (ANOVA *p* = 0.502), VCP-EC (ANOVA *p* = 0.756), AS-EO (ANOVA *p* = 0.426), and AS-EC (ANOVA *p* = 0.418).

To further investigate the potential relationship between COP fluctuations and athletes’ performance level, we analyzed the two marginal groups: (1) with the highest (Shooting) and (2) with the lowest (Figure Skating) postural stability.

In the Shooting group (Biathlon (*n* = 46), Practical Shooting (*n* = 10), Archery (*n* = 7), and Trapshooting (*n* = 9)), sports qualification ranks were available for 66 athletes. All the athletes were divided into 3 groups: (1) the low qualification group included First- and Second-Class Adult Sportsmen (LoQ, *n* = 31); (2) the medium qualification group included Candidates Master of Sports (MeQ, *n* = 28); (3) the high qualification group included Masters of Sports (HiQ, *n* = 7). The groups did not differ significantly in mean body weight (ANOVA F(2, 63) *p* = 0.35) and height (ANOVA F(2, 63), *p* = 0.29), although the mean age in the HiQ group (27.7 ± 8.6 years) was higher than in the LoQ and MeQ groups (16.2 ± 3.7 and 19.0 ± 7.7 years in the LoQ and MeQ groups, respectively. ANOVA F(2, 63) = 9.95, *p* = 0.00018; posthoc Tukey’s HSD test *p* < 0.01 for both groups). Athlete performance level had some effect on VCP-EO (ANOVA F(2, 63) = 4.19, *p* = 0.020; Figure 5). VCP-EO in HiQ was lower than in LoQ (Tukey’s HSD test *p* = 0.016), and on VCP-EC (ANOVA F(2, 63) = 2.90, *p* = 0.06): VCP-EC in HiQ was at a trend level lower than in LoQ (Post hoc Tukey’s HSD test *p* = 0.051). However, ANCOVA with age as the covariate invalidated the association of athletes’ performance level with VCP-EO (ANCOVA F(2, 62) = 2.17, *p* = 0.122). Thus, the association between performance level and VCP-EO was mostly due to the differences in age and/or longer sports experience.

In the Figure Skating group, athletes were divided into 4 qualification subgroups: (1) the low qualification group included First- and Second-Class Adult Sportsmen (LoQ, *n* = 28); (2) the medium qualification group included Candidates Master of Sports (MeQ, *n* = 34); (3) the high qualification group included Masters of Sports (HiQ, *n* = 12); (4) top qualification athletes, i.e., Masters of Sports who once held top places in the world figure skating ranking (TopQ, *n* = 6). The mean age in the LoQ group was less than in the other three groups (*p* < 0.001 vs. MeQ, HiQ, TopQ), the mean height and weight in TopQ were only less than in HiQ (*p* = 0.017). No significant intergroup differences in VCP-EO (ANOVA *p* = 0.318) and VCP-EC (ANOVA *p* = 0.408) were found, yet AS-EO (ANOVA, F(3, 76) = 2.35, *p* = 0.079; Figure 6A) and AS-EC (ANOVA F(3, 76) = 2.5, *p* = 0.065; Figure 6B) tended to increase in the TopQ group. Thus, according to the results of basic postural tests in a bipedal stance with EO and EC, COP fluctuations did not correlate with performance level in Figure Skating and showed a trend to increase in high-class athletes.

## 4. Discussion

The discussion of the study results is structured around the factors—age, sex, sports performance level, and shoe features—whose effects on postural stability in athletes we have studied.

### 4.1. Age-Related Differences in Postural Stability in Athletes

Regardless of the practiced sport, athletes at all ages showed higher postural stability than controls, based on individual stabilographic parameters (Table 2). In child and teenage athletes, postural stability in a bipedal stance was increased under EO conditions only: AS-EO and VCP-EO in athletes were lower than in control and remained so after age-based standardization. Our findings are consistent with earlier studies that have demonstrated a VCP decrease with age in healthy children [9,30,31]. Odenrick and Sandstedt [30] studied postural stability in 63 healthy children aged 3.5–17 years and showed that their postural sway amplitude decreased with age. Similar data were obtained for children of 3–11 years (n = 1181); the authors found that the rate of decrease in sway area and velocity gradually slowed down by the age of 10–11 [9,31]. The age-related increase in VCP and AS was probably associated with the maturation of the key components of the postural system (sensory, muscular, and central nervous systems [32,33,34]) and a better posture regulation due to a gradual shift of the COP from the heels forward, to the toes, with age [9,31]. It was shown that a functionally mature central nervous system fully integrates sensory information in order to optimize postural control. Normally, the CNS fully develops by the age of 9: the first to mature are sensory regions responsible for the use of visual information, followed by those responsible for proprioceptive information and then, finally, those responsible for the integration of vestibular information [34]. According to our data, postural stability increases, even until an older age of 18–30 years. Therefore, we believe that postural stability reaches its peak after adolescence, after the complete development of all body systems. In addition, systems involved in posture regulation develop unevenly, and their development may slow down during puberty [35]. The reduced VCP-EO and AS-EO in athletes show that practicing sports in childhood and adolescence boosts the development of postural stability, especially of systems responsible for the use of visual information to keep postural balance. However, it should be noted that the sports experience of child athletes was rather short, and their increased postural stability might have been due to their innate abilities for posture and movement regulation. Such children often go for sports and achieve high athletic results.

The leveling of differences in posture stability between adult athletes and controls under EO conditions was probably due to continued maturation of postural regulation in control subjects under the impact of environmental and natural factors.

At the age of 18 and older, however, higher postural stability in athletes versus controls was only observed under EC conditions: both VCP-EC and age-standardized VCP-EC in athletes were lower than in control (based on pairwise comparison, *p* = 0.002 and *p* = 0.003, respectively). The reduced VCP-EC is attributed to the more efficient use of proprioceptive and vestibular information [16]. These late differences between athletes and non-athletes are probably due to a later maturation of the respective sensory systems [34] and a favorable effect of sports training on the efficient use of proprioceptive information for balance control [16].

### 4.2. Sex-Related Differences in Postural Stability in Athletes

According to our data, postural stability (both under EO and EC conditions) in female athletes was higher than in male athletes, with the difference being most significant in the teenage (13–17 years) and adult (18 and older) subgroups. Reduced COP sway area and velocity in females were also noted by other authors [9,13]. Unlike our results, Nolan et al. (2005) found the highest postural stability in girls under 10 and failed to detect any sex-related differences in postural stability among healthy non-athletes at an older age of 12 or 16 [9]. In our study, the differences were more pronounced in teenage and adult athletes. According to yet another study [9], female alpine ski racers aged 14, 15, and 16 years demonstrated a better anterior–posterior balance on a moving platform than their male counterparts. At the age of 17–18 years, however, the lateral dynamic stability was higher in males. This is partly consistent with our data on the sex-related differences in postural control in teenagers and the beneficial effect of sports. Different authors link the higher postural stability of females with their (a) earlier physical maturation [9], (b) greater diligence and attention when doing postural tasks [11], (c) lesser body weight [12], (d) anatomical features: a lower center of gravity in teenage girls due to a relatively wider pelvis and narrower shoulders [9], (e) better trainability of postural regulation system [13]. Generally, the sex-related differences in postural stability can be probably explained by changes in the body habitus associated with sexual maturation: as opposed to the heavier lower body in adolescent females, the increasing weight of the upper body in adolescent males contributes to a more cranial location of the center of mass and hence a greater moment of inertia in young men, which results in lower stability of vertical posture compared with females. In addition, most females have a greater proprioceptive acuity of the lower limbs due to a lesser absolute muscle mass and strength [8], and this may be important for balance. However, further research is needed for a better understanding of the association between female sex and postural stability in athletes.

### 4.3. The Effect of Athletic Shoes on Postural Stability

As shown by our findings, in a pooled group of athletes from all sports where they train in hard boots with tight ankle support (Biathlon, Alpine Skiing, Snowboarding, Figure Skating, Freestyle, Speed Skating, Hockey, Short-Track, Cross-Country Skiing), both VCP-EO and VCP EC were higher than in those sports where athletes train in short sneakers with elastic soles (Figure 4). These data are consistent with another study [23], in which highly qualified national level alpine skiers had a larger sway area when standing on a firm-surface rocking platform than did less qualified regional level athletes. A similar pattern was observed when COP sway variables were compared between athletes of different performance levels within the Figure Skating group (Figure 6): in the TopQ group, AS-EO and AS-EC tended to be higher than in the less-qualified MeQ group. Taken together, our data are indicative of a relative decrease in postural stability caused by long use of rigid athletic footwear with tight ankle support in the training process. On the one hand, tight ankle support improves the running speed by facilitating the transmission of muscle effort from the foot to a slippery support and increases the stability of the posture in the boots by enhancing the tactile sensitivity and expanding the base of support [36]. On the other hand, this can strongly limit the normal activity of postural muscles and even worsen the postural balance [37]. Apparently, prolonged use of boots that restrict ankle mobility can virtually change posture regulation: the usual activity of the postural muscles gastrocnemius medialis and vastus medialis is restricted in such footwear [23]. As a result, balance is regulated through other mechanisms [23], which do not provide postural stability but can rather reduce it under standard conditions of standing barefoot [23]. These results allow us to recommend mandatory inclusion of balance exercises on a movable support without shoes into the preliminary and recovery stages of sports training in order to maintain and restore normal mechanisms of postural regulation involving the muscles of the ankle joint.

### 4.4. Sports Performance Level and Postural Balance

According to our results, sports performance level was weakly associated with postural stability: no correlation between sports performance level and postural stability was found in any of the age subgroups (children, teenagers, and adults) within the pooled group of all athletes from different sports. As mentioned above, a more detailed analysis yielded a negative correlation in the Figure Skating group and a positive correlation in the Shooting group. In high-class shooters, VCP-EO (*p* = 0.016) and VCP-EC (*p* = 0.051) were lower than in low-class shooters. These data confirm (a) a nonspecific nature of postural control improvement in various sports and (b) that a significant correlation between sports performance level and postural stability is noted in those sports where high postural stability is a specific and trained ability.

In general, our data are consistent with multiple studies [6,16,21,22,38,39] that did not reveal any strong association between the COP sway during a normal bipedal stance and sports performance level. Moreover, no relationship was found between the performance level in gymnastics and motor learning ability [18]. It can be surmised that athletes in different sports develop specific postural mechanisms for maintaining their body balance under specific static and dynamic conditions, e.g., when moving, sliding, cycling, jumping, and taking punches and pushes from partners. Due to specific postural adaptations, the mechanisms do not show up much in a normal erect posture. In shooting sports, however, the ability to keep a stable body posture in a bipedal stance is one of the specific basic abilities critical to athletic performance [15,40].

The mechanism behind the postural “super-stability” in shooters is probably associated with the development of a motor skill for voluntary stabilization of erect posture. They are learning to “freeze” their posture when they are aiming before shooting [15,41]. According to Mononen et al., the velocity of postural sway correlates with the rifle stability during aiming and accounts for about 26% of shooting performance [41]. Better than other athletes, shooters are able to control their postural sway in a postural test when instructed to stand straight and not to move. This postural skill is probably better, the higher the shooter’s performance level. As a result, there is a detectable association between stabilographic parameters and performance levels. Thus, the COP sway test in a bipedal stance can be considered (1) as a conventional standard physiological test for assessing the level of development of the postural system and its components, and (2) as a test for the level of motor skill for voluntary postural stabilization. The high postural stability in shooters reflects not only an excellent development of their posture regulation system but, even more so, their ownership of this postural skill. Its moderate development in other sports is due to its lesser importance as a technical element. Although all humans own this postural skill, the level of ownership differs. The assumption of a voluntary ability for postural stabilization in shooters requires additional research.

**Limitations.** The findings of this study have to be seen in light of some limitations.

Issues with sample and selection: the groups of athletes were not fully matched with the group of non-athletes based on sex, age, and the number of subjects; this could have affected the intergroup differences and, therefore, the conclusions. Additionally, the Sports groups were not homogeneous by sport; although the sports within a Sports group were similar according to certain criteria, they still differed in other aspects, which blunted the specificity of sports-related effect on postural stability.

Limited access to data: we did not have full access to relevant information about the athletes (e.g., individual features of their musculoskeletal system, history of injuries, training periods).

## 5. Conclusions

According to our findings, postural stability in athletes was mostly influenced by the athlete‘s age and, to a lesser extent, their sex, performance level, and the features of their athletic shoes. In child and adolescent athletes, better postural control was mainly due to more efficient use of visual information, whereas in adult athletes, this was due to other mechanisms, as shown by tests with eyes closed. Postural stability in female athletes was better than in male athletes, although the same was not true for non-athletes. Only in shooters, postural stability correlated with sports performance level, because keeping stable balance during quiet standing is critical in this sport. Long use of rigid athletic shoes with stiff ankle support was associated with reduced postural stability when measured barefoot.

The results show that the current functional ability of the postural system to support balance points to some clinical abnormalities. It would be useful to evaluate postural balance in children focused on serious engagement in sports that require excellent postural regulation and stability for high performance. Further studies are needed to determine sports-specific values of sway variables to help trainers assess the postural stability of athletes, taking account of their age, sex, sports performance level, and athletic shoe features.

## Figures and Tables

**Figure 1 sports-08-00089-f001:**
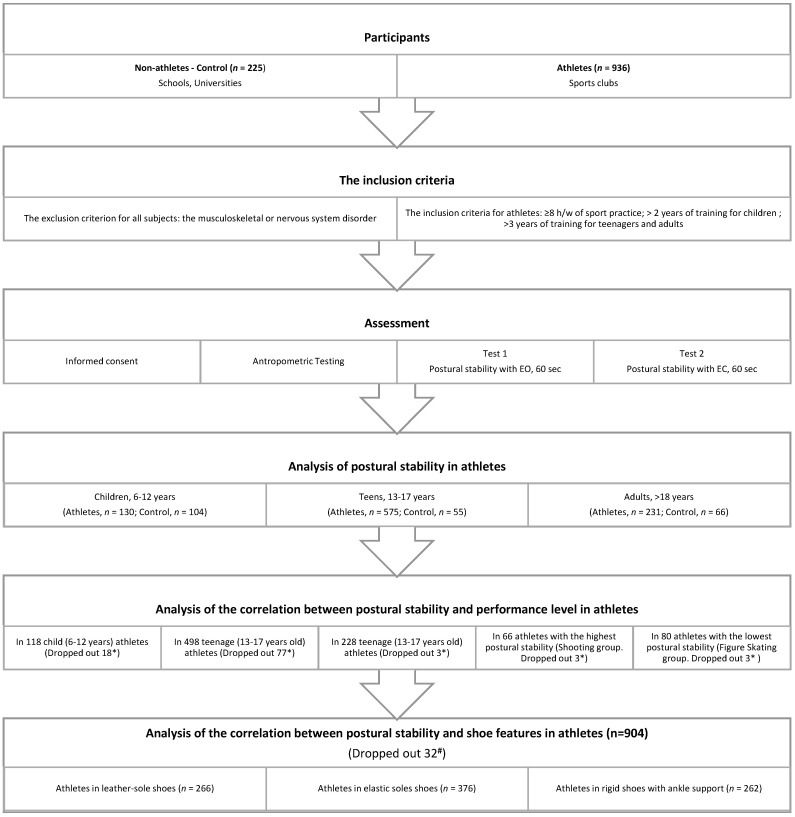
Flowchart of the study. Notes: *—Sports qualification data were lost or not provided; #—Rowing groups (*n* = 24) and sailing groups (*n* = 8) were not included in the analysis.

**Figure 2 sports-08-00089-f002:**
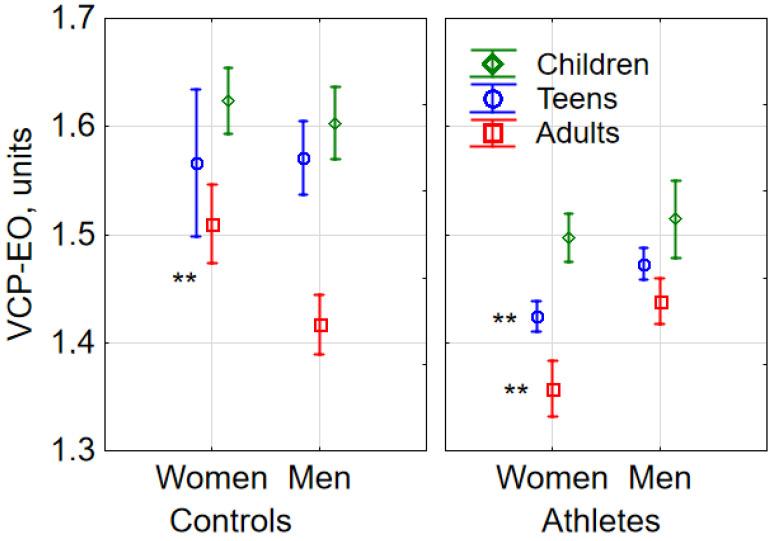
Sex-related differences in COP velocity under eyes-open (EO) conditions (VCP-EO) in athletes and controls (M ± 95% CI, where M is a mean and CI is a confidence interval). In the athlete group, VCP-EO in teenage and adult females was lower than in males in the respective age subgroups (** *p* < 0.01 vs. males, based on Tukey’s HSD test).

**Figure 3 sports-08-00089-f003:**
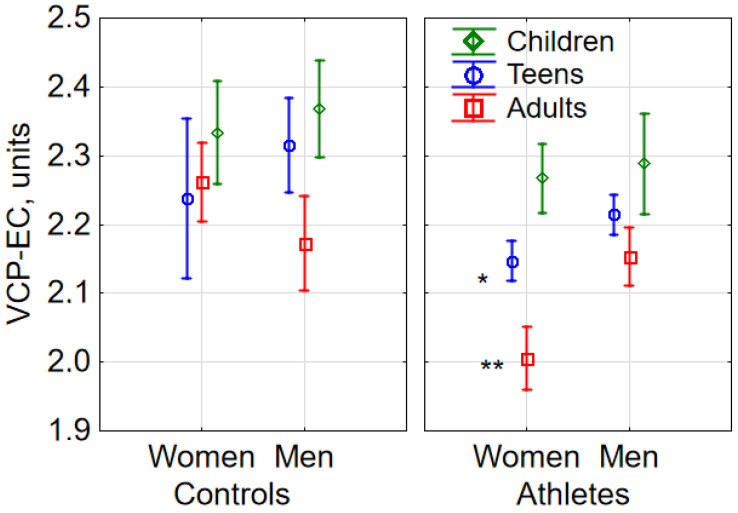
Sex-related differences in COP velocity under eyes-closed (EC) conditions (VCP-EC) in athletes and controls (M ± 95% CI). In the athlete group, in the teenage and adult subgroups, VCP-EC in females was lower than in males (* *p* < 0.05, ** *p* < 0.01, versus males, based on Tukey’s HSD test).

**Figure 4 sports-08-00089-f004:**
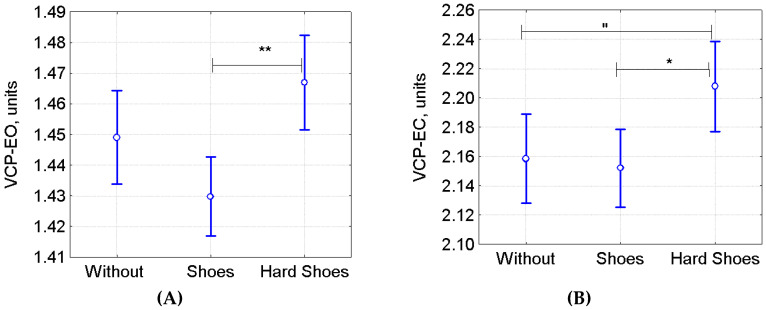
(**A**) VCP-EO; (**B**) VCP-EC into groups based on the type of athletic sports shoes (M ± 95% CI). *^,^** *p* < 0.001, “ *p* = 0.0066 based on Tukey’s HSD test.

**Figure 5 sports-08-00089-f005:**
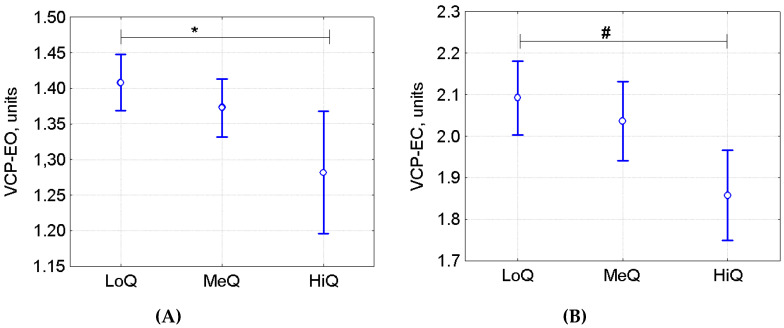
(**A**) VCP-EO (ANOVA *p* = 0.020); (**B**) VCP-EC (ANOVA *p* = 0.063) in the qualification-based subgroups of the Shooting group, (M ± 95% CI). LoQ—low qualification athletes, MeQ—medium qualification athletes, HiQ—high-qualification athletes, * *p* = 0.016, # *p* = 0.051 based on Tukey’s HSD test.

**Figure 6 sports-08-00089-f006:**
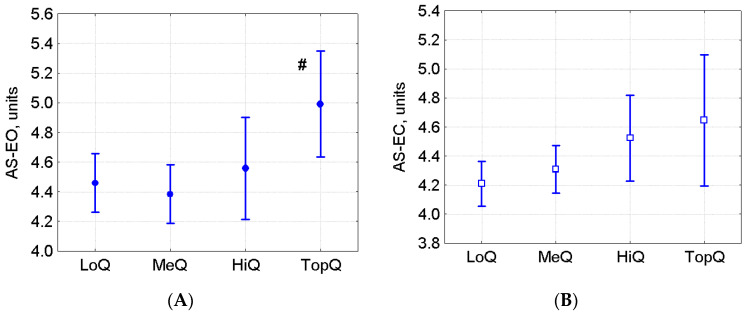
(**A**): AS-EO (ANOVA *p* = 0.079) and (**B**): AS-EC (ANOVA *p* = 0.065) in the Figure Skating group (M ± 95% CI). LoQ–low qualification athletes, MeQ–medium qualification athletes, HiQ–high qualification athletes, TopQ–top qualification athletes. # *p* < 0.054 vs. MeQ, based on Tukey’s HSD test.

**Table 1 sports-08-00089-t001:** Study groups.

No.	Group	Sport Disciplines (*n*)	F/M (*n*)	*n*
**1**	Team sports *	Basketball (*n* = 73), Handball (*n* = 17)	59/31	90
**2**	Shooting	Biathlon (*n* = 46), Practical shooting (*n* = 10), Archery (*n* = 7), Trapshooting (*n* = 9)	35/37	72
**3**	Boxing	Boxing (*n* = 38), Karate (*n* = 9), Kickboxing (*n* = 9), Thai Boxing (*n* = 10), Taekwondo (*n* = 40)	34/72	106
**4**	Tennis	Tennis (*n* = 43), Table tennis (*n* = 14)	25/32	57
**5**	Alpine Skiing	Alpine skiing (*n* = 32), Snowboarding (*n* = 17)	23/26	49
**6**	Figure Skating	Figure Skating (*n* = 80), Freestyle (*n* = 6)	59/27	86
**7**	Football	Football (*n* = 70)	36/34	70
**8**	Rowing	Rowing (*n* = 6), Canoeing (*n* = 14), Canoe slalom (*n* = 4)	6/18	24
**9**	Wrestling	Freestyle Wrestling (*n* = 8), Greco-Roman Wrestling (*n* = 29), Judo (*n* = 16), Sumo (*n* = 18)	18/53	71
**10**	Speed Skating	Speed Skating (*n* = 6), Curling (*n* = 17), Hockey (*n* = 18), Short Track (*n* = 20)	40/21	61
**11**	Cross-Country Skiing	Cross-Country Skiing (*n* = 37)	19/18	37
**12**	Running	Sprint Running (*n* = 4), Stayer Running (*n* = 7), Orienteering (*n* = 34)	20/25	45
**13**	Gymnastics	Artistic Gymnastics (*n* = 25), Rhythmic Gymnastics (*n* = 34), Cheerleading (*n* = 27), Trampoline Tumbling (*n* = 30), Climbing (*n* = 20), High Jumping (*n* = 10), Sailing (*n* = 8), Skeleton (*n* = 14)	110/58	168
**14**	Control	Non-athletes (*n* = 225)	97/128	225
	Total	Athletes (*n* = 936) Non-athletes (*n* = 225)	**581/580**	**1161**

Notes: * played with hands.

**Table 2 sports-08-00089-t002:** Stabilographic data by age group (mean (SD) of units).

Parameters	Children (6–12 Years, *n* = 234)		*t*-Test, *p*	Teens (13–17 Years, *n* = 630)		*t*-Test, *p*	Adults (18+ Years, *n* = 297)		*t*-Test, *p*
	Control	Athletes		Control	Athletes		Control	Athletes	
n	104	130		55	575		66	231	
Sex (F/M)	57/47	87/43	0.058 ^#^	20/35	301/274	0.024 ^#^	20/46	96/135	0.098 ^#^
Age, years	9.3 (1.7)	10.9 (1.3)	0.000	14.5 (1.5)	15.1 (1.3)	0.003	19.7 (2.3)	21.9 (4.9)	0.001
AS-EO, units	4.88 (0.53)	4.52 (0.55)	0.0001 *	4.68 (0.49)	4.45 (0.52)	0.001 *	4.32 (0.39)	4.32 (0.49)	0.973
VCP-EO, units	1.61 (0.12)	1.50 (0.11)	0.0001 *	1.57 (0.13)	1.45 (0.12)	0.001 *	1.45 (0.10)	1.40 (0.13)	0.020
AS-EC, units	4.46 (0.48)	4.32 (0.42)	0.019	4.32 (0.41)	4.32 (0.44)	0.919	4.19 (0.33)	4.19 (0.42)	0.968
VCP-EC, units	2.35 (0.26)	2.27 (0.23)	0.022	2.29 (0.25)	2.18 (0.22)	0.002 *	2.20 (0.21)	2.09 (0.25)	0.002 *

Notes: ^#^ based on the chi-squared test; * the differences between the parameters were significant (*p* < 0.05) according to ANCOVA. Age was used as the covariate.

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
