# Peer review of "Postural Stability in Athletes: The Role of Age, Sex, Performance Level, and Athlete Shoe Features"

_sports, 2020, doi:10.3390/sports8060089_

Round 1

Reviewer 1 Report

Interesting paper with an impressive amount of data.

I would suggest the following to improve the quality of the manuscript:

  1. English editing
  2. The rationale for the study is not clear. Please, develop the reason why this study was done. This is not really clear.
  3. Elaborate on the methods (please, be more specific regarding dependent & independent variables)
  4. Improve Table 2
  5. Figures: do not connect male and female mean results (lines are used only when there is a relationship)

Reviewer 2 Report

Comments to the Authors

Thank you so much for giving me the opportunity to review your study. I have given my comments and suggestions. Thank you.

Major comments

Please editing for English usage is needed throughout the manuscript.

Please organize the abstract once again.

Please change the citation order of references in the sentence according to the journal guidelines.

Please add the flowchart of this study.

Please describe tests and assessments in more detail.

Was the measurement sequence random?

It is necessary to explain the mechanism for easy understanding in the discussion.

Please reorganize concisely the clinical significance in the conclusion so that it is easy to understand.

There are many areas where the reference style does not fit the journal style at all.

Author Contributions also don't fit the journal style.

Minor comments

Please put tables and figures in the position in the sentence.

Unify terms “shoe features and expertise”, “shoe features and performance level”, “sport-related factors” to improve the readers' understanding.

Lines 49, 95 and 96: COP, VCP and AS, Avoid the first occurrence of an abbreviation in a sentence, because the abstract and the text are separate.

Line 71: Was the experiment measured by the same person during this period?

Line 75: Write down the approval number of IRB

Line 76: Why are the number of subjects in the experimental group and control group different?

Lines 88-90: What are the references for this statement?

Line 97: What are the references for this statement?

Line 208: Please add definitions on the shooting group.

Line 303: skating group and shooting group

Please add the limitations of this study.

Reviewer 3 Report

Line 24, might want to use another term in addition to “balance” to make it more clear this is meant to be about true balance, not in some holistic health sense.

Line 26 change assessment to “assessed”

Somewhere in the abstract explain shoe features or approach it in some manner.

Line 35-36 the conclusion statement doesn’t seem to have bearing on the data that is presented.

Explain what is meant by “effective postural balance”

Line 59 correct the grammar with assessment

Line 186 fist should be changed to “first” I believe

Line 193 is the same mistake as listed above.

Line 209 the same issue again.

Line 224 the same issue again.

Line 264-266 any references to back this up?

Otherwise well done article, did you assess the athletes for previous injury status to see if that influenced any of these postural values?

Round 2

Reviewer 2 Report

Comments to the Authors

Thank you so much for giving me the opportunity to review. The authors have improved the manuscript since the previous version. However, there are still a couple of minor issues. I have given my comments and suggestions. I hope this is clear and the authors might benefit from it.

Major comments

Please change the flowchart of this study. Write dropout rate and the order of evaluation method and so on.

Why wasn't measurement random?

All studies were performed in 2014-2018. Is “Protocol 11 dated 92 December 9, 2019” the approval number of IRB?

Minor comments

Please transfer the limitations the end of discussion.

Please reorganize the limitations to understand.

What does bold type mean?

Author Response

Dear colleague thank you SO MUCH for your attention for our work

This is very important for me

Be healthy and best regards

This experience takes our work to a whole new level thanks to you

Albina
